# ADAPTING WORLD MODELS WITH LATENT-STATE DYNAMICS RESIDUALS

## ABSTRACT

Simulation-to-reality reinforcement learning (RL) faces the challenge of reconciling discrepancies between simulated and real-world dynamics, which can degrade agent performance. When real data is scarce, a promising approach involves learning corrections to simulator forward dynamics represented as a residual error function, however this operation is impractical with high-dimensional states such as images. To overcome this, we propose ReDRAW, a latent-state autoregressive world model pretrained in simulation and calibrated to a target environment through residual corrections of latent-state dynamics rather than of explicit observed states. Using this adapted world model, ReDRAW enables RL agents to be optimized with imagined rollouts under corrected dynamics and then deployed in the real world. In multiple vision-based DeepMind Control Suite domains and a physical robot visual lane-following task, ReDRAW effectively models changes to dynamics and avoids overfitting in low data regimes where traditional transfer methods fail.

## 1 INTRODUCTION

Training robot control policies with reinforcement learning (RL) in real-world environments is inherently expensive, time-consuming, and risky because it requires extensive interactions with physical systems. Simulation provides a promising alternative as it offers a controlled, cost-effective, and parallelizable setting for generating data and training capable policies. However, leveraging simulated environments effectively is challenging due to inaccuracies in their representation of agent observations and dynamics. These inaccuracies create a sim-to-real gap, where simulated environments fail to correctly capture every relevant detail of real-world physics. This gap arises when real-world dynamics are only partially understood or are too expensive to model accurately. As a result, agents trained in simulation often struggle to successfully transfer their policies directly to real-world settings without additional adaptation [20].

One approach to addressing this gap is to use a small amount of real-world data to learn corrections to simulated transition functions, known as *residual dynamics corrections*. These corrections adjust the simulated dynamics to better match the real world, allowing for more accurate training of control policies [20; 35; 10]. However, this approach relies on the ability to efficiently learn corrections, which is difficult when the state information is represented in high-dimensional formats such as images. In these cases, significant feature engineering is often required to extract compact and meaningful state representations for learning residuals.

This work introduces a novel method for learning residual dynamics corrections directly in the *latent state space* of learned world models, eliminating the need for explicit feature engineering. Specifically, we build on latent-state world models such as Dreamer [13; 14; 15] that encode high-dimensional observations into compact latent states. These latent states can then be used to predict future dynamics, rewards, policy values, and optimal actions. World models enable RL agents to gather experience using synthetic trajectories in latent space, significantly reducing the need for real-world interactions.

Focusing on fully-observable robot domains, we propose a Markov Decision Process (MDP) world-model architecture, **DRAW** (**D**ynamics-**R**esidual **A**daptable **W**orld model), that encodes observations solely into a discrete latent state representation that better supports data-efficient transfer learning. After pretraining DRAW on simulated data, its weights are frozen to provide a fixed latent-state space during adaptation. A small offline dataset of real-world trajectories is then used to learn a residual

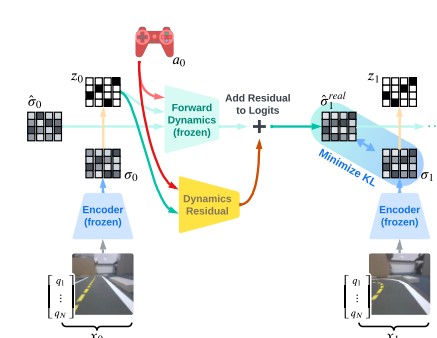

**Step 1. Pretrain World Model in Simulation**     **Step 2. Train Dynamics Residual with Real Data**

Figure 1: (**Left**) The DRAW world model is trained to encode states into a discrete latent representation without additional components, from which states, rewards, terminations, and future latent states are predicted. An RL agent can be trained in the world model via synthetic rollouts. (**Right**) The DRAW world model is frozen. Using a small reward-free dataset, world model dynamics are calibrated to a target environment by training an added residual error correction on latent state dynamics predictions. The RL agent can then be trained under rectified dynamics.

function in this fixed latent space. This function corrects the world model's dynamics, enabling it to more accurately represent real-world behavior. We refer to this residual-calibrated model as **Re**ctified DRAW (**ReDRAW**). RL agents can be trained with ReDRAW using imagined rollouts, producing policies that perform well in the real environment. Importantly, we do not require reward labels from the real environment to make this calibration, extending ReDRAW's applicability to real scenarios where rewards can be difficult to measure.

We evaluate ReDRAW on four vision-based DeepMind Control Suite (DMC) environments and further demonstrate the real-world usability of ReDRAW in sim-to-real applications by adapting from simulation to a physical real-time visual-navigation task on a Duckiebot robot [33]. Our experimental results suggest that ReDRAW outperforms traditional transfer learning methods in small data regimes to adapt to mismatched dynamics and avoids overfitting without early stopping. In real robot experiments, ReDRAW successfully performs simulation-to-reality dynamics adaptation with only 10K real steps ($\sim$17-minute demonstration), transferring from simulation with synthetic visual inputs to real-world images collected on the robot.

### CONTRIBUTIONS

1. We propose a new world-model architecture for dynamics adaptation in fully observable visual-control domains. DRAW encodes all state information into a single discrete latent space suitable for transfer in low-data regimes.

2. We demonstrate that the ReDRAW architectural extension can learn residual corrections in the latent space of DRAW to efficiently transfer between domains with mismatched dynamics, using only a small amount of offline target-domain data without reward labels.

3. We show that our method adapts dynamics from simulation to reality while also zero-shot transferring latent-state encoders from synthetic to real robot images.

4. Additionally, we open-source the code for our Unreal Engine [8] Duckiebot visual-control simulator to help facilitate further sim-to-real transfer research. Code and videos are available at https://redraw-research.github.io/project/.

## 2 PRELIMINARIES

### 2.1 RELATED WORK

Many existing methods calibrate or learn corrections to explicit state-transition models to better represent real dynamics during training [3; 2; 28], including through the use of error-correcting

residuals on simulation dynamics [20; 35; 10]. A limitation is that such explicit-state-based methods break down when all or part of the state representation is high-dimensional (e.g., images).

Latent-state world models like Dreamer [13; 14; 15] model dynamics and rewards for environments with high-dimensional input spaces in a condensed learned state representation, enabling sample-efficient training of RL agents within this compressed model of the environment. In Section 4.1.2, we show that Dreamer is prone to overfitting when pretrained on a source environment (a simulation) and finetuned on a small offline dataset of transitions from a target environment with modified dynamics (the real world). This is a major issue when real-world evaluation is logistically challenging and only doable in limited quantities.

In this work, for fully observable environments, we find that along with a few other architectural changes (Section 3.1, Appendix F.1), representing the latent state with only a discrete representation (as opposed to e.g., with a GRU state) allows the world model to be frozen after pretraining and its latent-state dynamics calibrated using an added error-correcting residual component. Compared to DreamerV3 [15], on the same offline datasets, we see a remarkably improved robustness to transfer-time overfitting with the proposed ReDRAW method.

Other approaches like physics domain randomization [34; 29; 7] and system-parameter identification [37; 1] use a configurable simulator along with expert knowledge of how simulated and real dynamics may differ to train agents that are robust to a variety of real physics. Often, a simulator cannot represent real dynamics under any parameterization, and differences between sim and real physics may not be known. For ReDRAW, we do not rely on configurable simulator physics or privileged insight into the discrepancies between environment dynamics. Offline RL techniques [25; 9; 26] can also learn policies from fixed real datasets, but they require real reward labels, which we assume unavailable since rewards can be challenging to measure in many real-world settings.

Due to the assumptions of states with high-dimensional image components, fixed simulator physics, and no real-reward labels, few existing works are meaningfully comparable in our setting. Experiments in this work primarily compare the ReDRAW adaptation method to other methods for fine-tuning a latent-state world model. In Appendix D, we also compare physics domain randomization on our robot sim-to-real task, which performs proportionally to how well the simulator can be configured to represent a distribution of potential real physics conditions. An extended related-work discussion is available in Appendix B, including a comparison of the high-level assumptions made by ReDRAW with other methods in Table 2.

## 2.2 PROBLEM DEFINITION

We consider two MDPs, denoted as $M_{\text{sim}}$ and $M_{\text{real}}$, which share the same state space, action space, and reward function, but differ in their transition dynamics. Formally, each MDP is defined by a tuple $M_i = (X, A, R, \gamma, P_i)$, with a shared state space $X$, action space $A$, reward function upon entering a state $R : X \to \mathbb{R}$, discount factor $\gamma \in [0, 1)$, and stochastic transition function $P_i$ for $i \in \{\text{sim}, \text{real}\}$.

Our objective is to find a policy $\pi_{\text{real}}$ that achieves high expected discounted cumulative reward in $M_{\text{real}}$, $J_{\pi,\text{real}} = \mathbb{E}_{\pi, P_{\text{real}}}[\sum_{t=0}^{\infty} \gamma^t R(x_t)]$. To capture logistic challenges common in real robot settings, we have access to a limited amount of offline reward-free data $(x_t, a_t, x_{t+1})$ from $M_{\text{real}}$. To make up for this, we can collect a large amount of online reward-labeled experience $(x_t, a_t, x_{t+1}, r_{t+1})$ in $M_{\text{sim}}$. Our method aims to produce a well-performing agent in $M_{\text{real}}$ by learning a compressed latent-state world model to emulate the simulation's dynamics and reward functions. We then calibrate this world model's latent dynamics on the limited real transition data such that a performant agent can be trained by collecting synthetic experience in the rectified world model.

Finally, sim-to-real transfer poses two challenges: adapting dynamics and transferring perception. Our proposed ReDRAW method addresses dynamics adaptation. For perception, we apply standard zero-shot techniques like image augmentation and camera-parameter randomization to ReDRAW and every baseline, outlined in Section 4.2.1.

## 3 METHOD

In this section, we describe our MDP world model architecture DRAW (Figure 1, Left) and its counterpart with calibrated dynamics, ReDRAW (Figure 1, Right). We first define the DRAW model,

how it represents latent states and dynamics, and how it is trained. Then we describe how we facilitate sample efficient transfer learning of dynamics by training a residual error correction on latent-state transitions, creating the ReDRAW world model.

### 3.1 DRAW ARCHITECTURE AND PRETRAINING

We use DRAW to model an MDP by encoding state inputs into a compressed stochastic latent representation using variational inference. Similar to DreamerV3 [15], our latent representation is trained via objectives for state and reward reconstruction along with future latent-state prediction. We then train an actor–critic reinforcement-learning agent on latent-state inputs by autoregressively rolling out synthetic trajectories as experience and using reconstructed rewards as a learning signal. Finally, the actor can be deployed to the environment by encoding immediate state inputs as latent states and providing these encodings to the actor. Figure 1 (Left) depicts connections during DRAW world model training, while Figure 5 in Appendix A depicts actor–critic training and deployment.

We model the latent state purely as a single stochastic multi-categorical discrete variable $z_t \in \mathcal{Z}$. $z_t$ is a $K$-tuple of conditionally independent categorical variables, each represented as a 1-hot vector of length $N$. We denote $z_t$ as the latent state encoded from the immediate state $x_t$ (1) and $\hat{z}_t$ as the latent state predicted via world model dynamics from the previous latent state and action (5). We denote $\hat{u}_t \in \mathbb{R}^{K \times N}$ as the logits for the multi-categorical distribution of $\hat{z}_t$ and $\hat{\sigma}_t = \text{softmax}(\hat{u}_t)$ as the $K$ concatenated normalized probability vectors. To estimate gradients in the sampling step for $z_t$ or $\hat{z}_t$, we use the straight-through estimator [4; 14].

By compressing all state information into a single discrete representation $z_t$, we aim to provide a well-structured encoding of the underlying state $x_t$, enabling the learning of generalizable functions, such as residual corrections, from limited data using $z_t$ as input. Illustrated in Figure 1 (Left), we define our DRAW world model and actor–critic agent, respectively parameterized by $\theta$ and $\phi$, as:

$$\begin{array}{llr}
\text{State Encoder} & z_t \sim q_\theta(z_t|x_t) & (1) \\
\text{Forward Dynamics} & \hat{u}_t = f_\theta(z_{t-1}, \hat{\sigma}_{t-1}, a_{t-1}) & (2) \\
\text{Forward Belief} & \hat{\sigma}_t = p_\theta(\hat{z}_t|z_{t-1}, \hat{\sigma}_{t-1}, a_{t-1}) & (3) \\
& = \text{softmax}(\hat{u}_t) & (4) \\
\text{Forward Sample} & \hat{z}_t \sim \text{MultiCategorical}(\hat{\sigma}_t) & (5)
\end{array}$$

$$\begin{array}{llr}
\text{Reward} & \hat{r}_t \sim p_\theta(\hat{r}_t|z_t) & (6) \\
\text{Continuation} & \hat{c}_t \sim p_\theta(\hat{c}_t|z_t) & (7) \\
\text{State Decoder} & \hat{x}_t \sim p_\theta(\hat{x}_t|z_t) & (8) \\
\text{Policy} & a_t \sim \pi_\phi(a_t|z_t) & (9) \\
\text{Value Function} & v_t = V_\phi(z_t) & (10)
\end{array}$$

We represent all functions in DRAW as multi-layer perceptrons (MLPs) except for image components of the state encoder and decoder, which are convolutional (CNNs). Interestingly, we found that providing $\hat{\sigma}_{t-1}$ as an input to the forward dynamics function $f_\theta$ significantly increased our downstream adaptation performance. We speculate that this is because $\hat{\sigma}_{t-1}$ helps provide a gradient signal for learning features relevant for long-term dynamics predictions without adding additional dimensionality to state prediction outputs. We provide ablations on this design choice in Appendix F.1.

We optimize DRAW on $M_{sim}$ with a prediction loss $\mathcal{L}_{\text{pred}}$ to reconstruct states, rewards, and episode terminations, as well as a dynamics loss $\mathcal{L}_{\text{dyn}}$ and a representation loss $\mathcal{L}_{\text{rep}}$ to learn latent-state dynamics under a predictable representation. Drawing subtrajectories $\zeta$ from a buffer of interaction experience, the world-model loss function $\mathcal{L}(\theta)$ is:

$$\mathcal{L}(\theta) = \mathbb{E}_{q_\theta(z_{1:T}|\zeta)} \left[ \sum_{t=1}^{T} \beta_{\text{pred}} \mathcal{L}_{\text{pred}}^t(\theta) + \beta_{\text{dyn}} \mathcal{L}_{\text{dyn}}^t(\theta) + \beta_{\text{rep}} \mathcal{L}_{\text{rep}}^t(\theta) \right], \quad (11)$$

where $T$ is the length of $\zeta$, and for $t = 1, \ldots, T$:

$$\mathcal{L}_{\text{pred}}^t(\theta) \doteq -\ln p_\theta(x_t|z_t) - \ln p_\theta(r_t|z_t) - \ln p_\theta(c_t|z_t) \quad (12)$$

$$\mathcal{L}_{\text{dyn}}^t(\theta) \doteq [\mathbb{D}[q_{\bar{\theta}}(z_t|x_t)||p_\theta(\hat{z}_t|z_{t-1}, \hat{\sigma}_{t-1}, a_{t-1})]]_1 \quad (13)$$

$$\mathcal{L}_{\text{rep}}^t(\theta) \doteq [\mathbb{D}[q_\theta(z_t|x_t)||p_{\bar{\theta}}(\hat{z}_t|z_{t-1}, \hat{\sigma}_{t-1}, a_{t-1}))]]_1, \quad (14)$$

with $\mathbb{D}$ the Kullback–Leibler divergence, $[\cdot]_1$ denoting clipping to 1 any value below 1, corresponding to free bits [23], and $\bar{\theta}$ a stopped-gradient copy of $\theta$.

We train the actor–critic agent with the same procedure and losses as DreamerV3, providing the DRAW world model state $\hat{z}_t$ as agent inputs during imagined rollouts and $z_t$ during data collection

and evaluation. When training the actor–critic, we seed synthetic rollouts with starting states $x_0$ drawn from the same experience buffer as used for world-model training. We do not backpropagate value gradients through dynamics, and we train the policy using the Reinforce objective [41] with normalized returns and critic baselines [15].

We alternate mini-batch updates between the world model and the actor–critic. During source environment pretraining, updates are interleaved with online data collection. Since we cannot fully predict which trajectories in $M_{sim}$ will best facilitate learning transferable features and dynamics for $M_{real}$, we employ Plan2Explore [36] to provide intrinsically motivated exploration, encouraging the collection of a highly diverse set of source-environment trajectories.

### 3.2 ADAPTATION VIA LATENT DYNAMICS RESIDUALS

After pretraining the DRAW world model online in the $M_{sim}$ environment with a large amount of data, we propose the ReDRAW architecture and method to use a small offline dataset of transitions from the target environment to calibrate DRAW's dynamics to match $M_{real}$ using a latent-state error residual.

We model the dynamics residual using an MLP $\delta_\psi$ that predicts a correction $\hat{e}_t$ to the forward-dynamics logit vector $\hat{u}_t$. This correction produces a modified transition distribution $\hat{\sigma}_t^{real}$, from which forward latent-state predictions $\hat{z}_t^{real}$ are sampled to approximate $M_{real}$. We formulate the calibrated dynamics as:

$$\hat{u}_t = f_\theta(z_{t-1}, \hat{\sigma}_{t-1}^{real}, a_{t-1}) \quad (15) \qquad \hat{\sigma}_t^{real} = p_{\theta,\psi}(\hat{z}_t^{real}|z_{t-1}, \hat{\sigma}_{t-1}^{real}, a_{t-1}) \quad (17)$$

$$\hat{e}_t = \delta_\psi(z_{t-1}, a_{t-1}) \quad (16) \qquad = \mathrm{softmax}(\hat{u}_t + \hat{e}_t) \quad (18)$$

$$\hat{z}_t^{real} \sim \mathrm{MultiCategorical}(\hat{\sigma}_t^{real}). \quad (19)$$

To train the residual on real data, we freeze the world-model weights $\theta$ and only optimize the parameters $\psi$ of the residual network $\delta_\psi$. In the transfer phase, we only optimize the actor–critic agent and a new loss $\mathcal{L}_\delta(\psi)$ on the rectified world-model dynamics. Our objective is to predict corrections $\hat{e}_t$ of $\hat{u}_t$ so that our new dynamics predictions $\hat{\sigma}_t^{real}$ match the observed encoder distribution over latent states collected in $M_{real}$. The loss function for the residual is:

$$\mathcal{L}_\delta(\psi) = \mathbb{E}_{q_{\bar{\theta}}(z_{1:T}|\zeta^{real})} \left[ \sum_{t=1}^{T} \mathbb{D}[q_{\bar{\theta}}(z_t|x_t)||p_{\bar{\theta},\psi}(\hat{z}_t^{real}|z_{t-1}, \hat{\sigma}_{t-1}^{real}, a_{t-1})] \right]. \quad (20)$$

Since we consider fully observable environments, the target encoder latent-state distribution $q_\theta(z_t|x_t)$ depends solely on $x_t$ and can be frozen after pretraining in $M_{sim}$ if the collected source-environment data adequately covers the state space. As a result, the latent-state representation for ReDRAW in $M_{real}$ is unchanged from DRAW in $M_{sim}$.

Notably, due to this unchanged latent-state representation between $M_{sim}$ and $M_{real}$, the frozen DRAW $M_{sim}$ reward function $p_\theta(\hat{r}_t|\hat{z}_t)$ can be reused in world-model rollouts to train the ReDRAW agent with $p_\theta(\hat{r}_t|\hat{z}_t^{real})$ in $M_{real}$, eliminating the need for reward data from $M_{real}$. This is particularly beneficial since building a reward recording system in real-world scenarios, such as robotics, often requires costly and complex setups like additional sensors or feedback mechanisms, which may be infeasible in certain environments.

Finally, given the ReDRAW world model with dynamics adapted to match $M_{real}$, the actor–critic can learn a high-performing policy for the new environment by training in the world model under the new rectified dynamics, using $\hat{z}_t^{real}$ as input during training. In our experiments, we alternate agent and world-model training during adaptation to measure the agent's performance as world model training progresses.

## 4 EXPERIMENTS

We evaluate ReDRAW in two distinct settings: (1) adapting from DeepMind Control (DMC) [38] environments to modified counterparts with changed physics, and (2) transferring from a simulation

in Unreal Engine to a real robot visual lane-following task using the Duckietown [33] platform. Our experiments address three main questions:

1. How do latent-space residuals compare to traditional finetuning methods in correcting world-model dynamics under limited target-domain data?

2. How do data quantity and collection policies influence transfer performance?

3. Can ReDRAW effectively close the sim-to-real gap in a robotics task with visual inputs?

## 4.1 DeepMind Control Experiments

### 4.1.1 DMC Domains

We first consider four pairs of source and target environments from the DMC suite, each pair having the same state and reward structure but mismatched dynamics. We use original environments from DMC as sources, while the target environments introduce physics modifications such as applied wind, external torque, or reversed actions. For a detailed description, refer to Appendix I. These differences in dynamics between source and target environments are substantial enough to require policy adaptation for optimal performance. Although dynamics differ between source and target, the state spaces, reward functions, and episode termination conditions remain unchanged. To maintain full observability, we represent the state as an image paired with a vector of joint velocities. In Appendix H, we also demonstrate comparable performance using framestacking for the same purpose.

To pretrain on each source environment, we collect 9 million environment steps (4.5e6 decision steps with an action repeat of 2) using Plan2Explore [36], which promotes diverse state visitation rather than narrowly exploiting the original environment's reward function. After this phase, we adapt to each target environment using a small offline dataset of 40K decision steps (equivalent to 80 episodes), gathered by an expert policy in the target domain.

### 4.1.2 Comparison with Finetuning

We compare ReDRAW with several baselines that attempt to adapt a pretrained world model to the new domain. Critically, except where noted with *, the methods we test do not use reward labels or train with a reward-reconstruction objective during the adaptation phase. These baselines include:

**DRAW/DreamerV3 Zeroshot:** We take the source-trained DRAW or DreamerV3 agent and deploy it in the target environment without any adaptation.

**DRAW/DreamerV3 Finetune:** The world model and agent are finetuned on the target-domain offline dataset. To mitigate overfitting on the small dataset, we freeze the world model encoder and decoder parameters and only retrain the agent and dynamics components. For DRAW this entails optimizing only $f_\theta$ with $\mathcal{L}_{\text{dyn}}$. Analogously, for DreamerV3, the RSSM recurrent prior and posterior components are updated while leaving the observation feature embeddings and decoders unchanged.

**DRAW/DreamerV3 Finetune (No Freeze)*:** Every component, including the encoder and decoder, is finetuned using all original world model loss terms. These are the only two baselines requiring access to reward data during adaptation.

**DRAW New Dyn:** The entire world model is frozen after pretraining, but instead of learning a residual addition to $f_\theta$, we train a new dynamics function $\hat{\sigma}_t^{real} = g_\psi(\hat{\sigma}_t, z_{t-1}, a_{t-1})$ with a similar capacity to $\delta_\psi$ and conditioned on the next-latent-state distribution $\hat{\sigma}_t$ predicted by the frozen source dynamics (Eq. 3). This method demonstrates an alternate way to leverage frozen dynamics predictions learned from the source environment. Other variations of this baseline are investigated in Appendix G.

Figure 2 shows the returns in each target domain as a function of offline updates on each target dataset. Zero-shot deployment without adaptation fares poorly in these altered dynamics. Finetuning approaches initially improve in some cases but all eventually overfit to the small dataset. The *No Freeze\** variations are quicker to overfit than their partially frozen counterparts.

In contrast, ReDRAW's latent-space residual method attains a sustained level of high-performance and avoids overfitting during the 3 million updates (1-3 days of training) we test on. This highlights a critical benefit of the ReDRAW transfer method: once ReDRAW reaches high performance in the target domain, it demonstrates a remarkable resistance to performance degradation. ReDRAW's

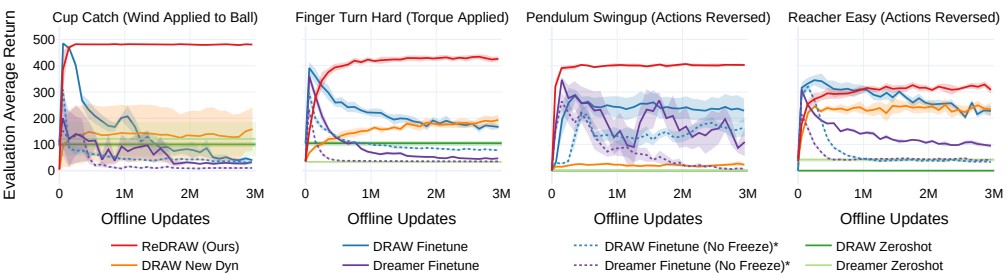

Figure 2: Average evaluation episode return transferring from each DMC environment to a modified variant of it given 40K offline target environment transition samples. Shaded regions indicate the standard error of the mean over 4 seeds for each method. ReDRAW consistently achieves high returns in the target environments and avoids overfitting.

ability to avoid overfitting for long periods of time makes it highly applicable to sim-to-real scenarios where validation testing on a real robot often cannot practically be done repeatedly and educated guesses need to be reliably made regarding stopping conditions of the training process.

ReDRAW excels at maintaining a high degree of validation performance by preserving existing dynamics predictions learned in simulation where data is abundant and using the limited target data to learn a low-complexity adjustment to those predictions. Comparing ReDRAW with *DRAW New Dyn*, we see that while both approaches utilize both the previous state and the frozen simulation dynamics predictions, the residual operation appears to play a key role in limiting the complexity of the changes made to the original world model dynamics, allowing ReDRAW to avoid overfitting.

### 4.1.3 DATA POLICIES AND QUANTITY

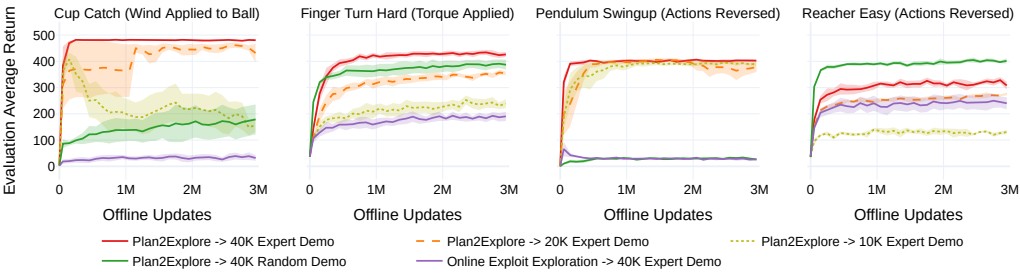

Figure 3: Impact of offline adaptation dataset size and source/target domain data collection strategies on ReDRAW. Expert demonstrations consistently provide useful target domain data for adaptation. Collecting diverse simulation experience with a method like Plan2Explore is essential for good transfer performance.

Figure 3 examines the effect of data collection on ReDRAW's transfer performance by ablating: (i) the diversity of source-domain experience, contrasting Plan2Explore simulation data collection with the exploit policy, and (ii) the quality and quantity of target-domain transitions, comparing expert demonstrations versus random actions and varying expert dataset sizes (40K, 20K, 10K). In our default configuration (*Plan2Explore → 40K Expert Demo*), ReDRAW attains the strongest transfer performance.

**Source-Domain Data:** Collecting source trajectories with Plan2Explore consistently yields high returns after adaptation. When we replace Plan2Explore with the exploit policy as the pretraining data collection policy, critical source transitions that may help in the prediction of the target dynamics are missed and transfer performance is reduced significantly. This demonstrates that exploratory breadth in simulation rather than narrowly optimizing the source reward is essential for learning latent features that are useful for downstream residual corrections to match altered physics.

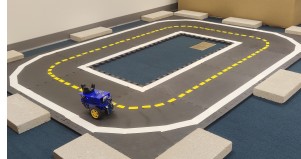 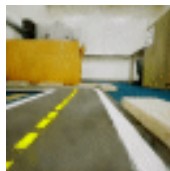 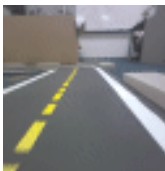

(a) Digital Twin Simulation     (b) Real Robot Environment     (c) Sim State Image (d) Real State Image

Figure 4: (a) Digital-twin simulation constructed using Gaussian splatting [21]. (b) Real-world robot lane-following environment. (c) Simulation state image component. (d) Real-world state image component. The agent is tasked to drive quickly around the track while staying near the lane center using an egocentric camera and velocity sensor. We train our DRAW world model in simulation and calibrate its dynamics with ReDRAW on a small dataset of human demonstrations with mixed optimality, producing a successful agent in the real environment.

**Target-Domain Data:** Adapting with expert demonstrations consistently yields effective transfer, while using a dataset of random actions can either help or hurt transfer performance in comparison. We speculate that expert data is useful for modeling pertinent dynamics for performing well while high-entropy actions may additionally be useful in modeling failure scenarios for the agent to avoid. Investigating the quantity of target-domain data, reducing the size of the expert dataset leads to a drop in average return with 20K samples and a further drop along with overfitting in Cup Catch with 10K samples. Taken together, these results indicate a practical lower bound on required expert data for robust transfer with ReDRAW.

## 4.2 DUCKIEBOT SIM-TO-REAL TRANSFER

Finally, we evaluate ReDRAW in a sim-to-real robotic lane-following task using the Duckietown platform [33]. Here, the agent controls a wheeled robot to navigate around a track while remaining centered in its lane. The state space includes a forward-facing camera image plus egocentric forward and yaw velocity values, and actions are defined as continuous forward and yaw target velocities in $[-1, 1]$. To provide a simulation to transfer from, we construct an environment in Unreal Engine using a Gaussian splat [21] reconstruction of the robot's environment to mimic the robot's state space. Figures 4a and 4b show the digital twin and real environment, respectively. We also implement the simulation with a rough approximation of real dynamics, although details like precise handling while driving and control rate (6Hz sim vs 10Hz real) still differ from the real robot.

The Duckiebot receives rewards proportional to its projected velocity along the lane-center path but instead incurs penalties when it deviates too far from this path. When moving forward, we also penalize the agent proportionally to its yaw velocity to encourage smooth driving. Simulation episodes terminate either when the robot leaves the track, with a large penalty applied, or after 200 steps. Exact experiment details are presented in Appendix C.

### 4.2.1 BRIDGING THE SIM-TO-REAL VISION GAP

Despite efforts to recreate the real environment, visual disparities between the simulation and real environment still exist (Figure 4c vs 4d). Although our main focus in this paper is adapting dynamics, we employ visual randomization [40] along with image augmentation [5] for all compared methods to bridge the sim-to-real vision gap. Each episode, we randomize the simulation camera's mounted location on the robot, camera tilt, and its field of view. We also apply image augmentations at train time to both sim and real image inputs to learn world-model image encoders robust to task-irrelevant features like lighting, color hue, and background furniture placement. Similar to [22], we train with augmented inputs, but the world-model decoder still reconstructs the original images as targets in $\mathcal{L}_{\text{pred}}$, thus focusing the latent-space features on task-relevant elements rather than the irrelevant augmentations we apply. We apply this asymmetric decoding objective to both DRAW and DreamerV3. ReDRAW trains its residual with augmented inputs but has no decoding objective during transfer learning since its latent-state representation is fixed.

Table 1: Mean and SEM performance on the real Duckiebots lane-following task aggregated over 5 episodes each for 4 training seeds. Agents are given 300 steps (30 seconds) to complete a lap from a fixed starting position. *Center Offset* denotes distance from the lane center. Absence of a *Lap Time* indicates all runs either failing to complete a lap or terminating early by driving off the track.

| Method | Transfer Sim to Unmodified Real | | | Transfer Sim to Actions-Reversed Real | | |
| | Avg Dense Reward ($\uparrow$) | Avg Lap Time (sec) ($\downarrow$) | Avg Center Offset ($\downarrow$) | Avg Dense Reward ($\uparrow$) | Avg Lap Time (sec) ($\downarrow$) | Avg Center Offset ($\downarrow$) |
|---|---|---|---|---|---|---|
| Dreamer Zeroshot | -1.18 $\pm$ 0.23 | – | 6.86 $\pm$ 0.56 | -2.35 $\pm$ 0.23 | – | 13.36 $\pm$ 1.13 |
| Dreamer Finetune | -0.87 $\pm$ 0.33 | – | 5.45 $\pm$ 1.55 | -1.61 $\pm$ 0.57 | – | 7.75 $\pm$ 1.53 |
| DRAW Zeroshot | 0.07 $\pm$ 0.06 | **22.41** $\pm$ 0.73 | 5.12 $\pm$ 0.41 | -2.72 $\pm$ 0.42 | – | 9.39 $\pm$ 1.35 |
| ReDRAW (Ours) | **0.38** $\pm$ 0.02 | **22.75** $\pm$ 0.26 | **2.47** $\pm$ 0.26 | **0.39** $\pm$ 0.03 | **24.21** $\pm$ 1.15 | **2.10** $\pm$ 0.39 |

### 4.2.2 TRANSFERRING TO THE REAL ROBOT

We pretrain DRAW and DreamerV3 in simulation using 600K random actions followed by 1.4 million online steps with Plan2Explore. On the real robot, we collect a small offline adaptation dataset of 1e4 timesteps ($\sim$17 minutes) using human demonstrations employing a mixture of safe random actions and semi-proficient driving. Table 1 compares performance using this offline dataset to adapt to two variations of the real environment, *unmodified real*, where minor physics disparities between sim and real are the natural result of inaccurate dynamics modeling, and *actions-reversed real*, where actions (in adaptation data and deployment) are inverted, requiring large but regular adaptation to drive successfully. We adapt ReDRAW and DreamerV3 *Finetune* for 2e5 offline updates. All methods are evaluated for 5 episodes each over 4 training seeds. We additionally compare to physics domain randomization, in which the world models and agents are trained with a randomized physics training regime that other methods do not have access to, in Appendix D.

In *unmodified real*, DRAW *zeroshot* is able to successfully drive despite never seeing real data but incurs low rewards by veering far from the lane center. DreamerV3 *zeroshot* fails, driving off the track in all lap attempts. We speculate that DreamerV3 *zeroshot* fails while DRAW *zeroshot* succeeds because DreamerV3's recurrent model observes out-of-distribution sequences under changed dynamics, resulting in inaccurate latent-state predictions. DRAW and ReDRAW are non-recurrent in deployment time and cannot suffer from this same issue. Similar to DMC experiments, DreamerV3 *Finetune* fails to adapt, possibly due to overfitting, and ReDRAW achieves significantly higher average dense rewards than DRAW *zeroshot* by training with corrected dynamics and staying close to the lane center.

In the more extreme *actions-reversed real* transfer task, ReDRAW is the only method that successfully adapts and completes laps on the real robot due to the incompatibility of zero-shot policies to this environment and the limited real data in the case of DreamerV3 *Finetune*. These results demonstrate that ReDRAW can be effectively used to adapt dynamics from simulation to reality using a limited offline real dataset without rewards, and that ReDRAW can be combined with visual adaptation methods to do so.

## 5 LIMITATIONS AND FUTURE WORK

A potential limitation of ReDRAW is that it maintains high target-environment performance over many updates and avoids overfitting in-part due to the low complexity of the residual component. This suggests that only conceptually simple changes to dynamics may effectively be modeled with low amounts of data, warranting future investigation. Our method for collecting a broad set of simulation training data via Plan2Explore is also heuristic. We would like to investigate the problem of guiding simulation data collection for better transferability by adopting exploration policies informed by the offline target-environment data. Finally, we want to explore if residual adaptation methods can be meaningfully applied to foundation world models, efficiently converting them from generators of plausible dynamics to generators of specific dynamics.

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

# A    ACTOR-CRITIC TRAINING AND DEPLOYMENT

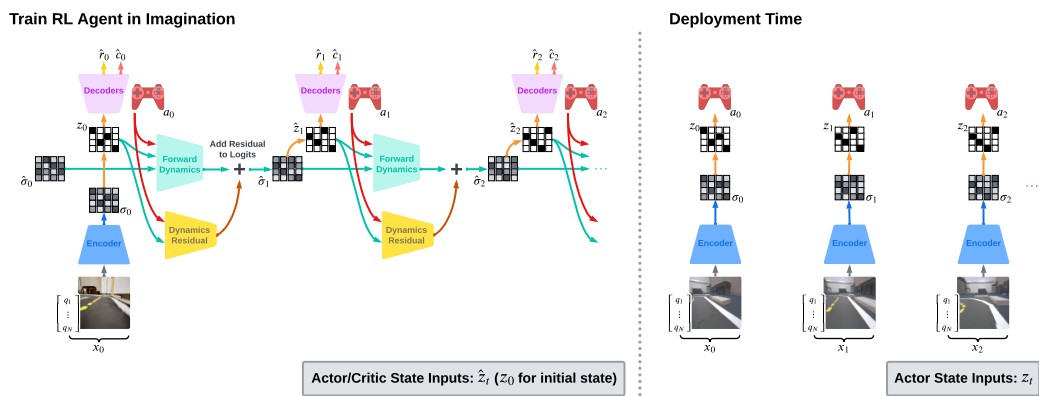

Figure 5: (**Left**) The actor and critic are trained by interacting with the world model. Starting from an environment state sampled from the replay buffer, the world model generates imagined rollouts using actions provided by the actor. The residual component is omitted during DRAW pretraining. (**Right**) At deployment, only the encoder and actor modules are utilized. The immediate environment state is processed by the encoder, and the actor generates an action based on $z_t$ sampled from $\sigma_t$.

# B    EXTENDED RELATED WORK

Table 2: Comparison of alternative methods against key desiderata. ReDRAW uses a small reward-free real-world dataset to calibrate a learned dynamics model of a simulation and match the real environment. ReDRAW uses a latent-state representation to maintain compatibility with high-dimensional state components like images, and it is agnostic to the types of discrepancies between simulation and real dynamics.

| Method | No Real Rewards | High-Dimensional Image Inputs | No Configurable Simulator or Disparity Insight | Low Real Data |
|---|:---:|:---:|:---:|:---:|
| ReDRAW (Ours) | ✓ | ✓ | ✓ | ✓ |
| World Model Finetuning | ✓ | ✓ | ✓ | ✗ |
| Adapting Explicit State Transitions | ✓ | ✗ | ✓ | ✓ |
| Offline RL | ✗ | ✓ | ✓ | ✓ |
| Physics Domain Randomization | ✓ | ✓ | ✗ | ✓ |

This research lies at the intersection of sim-to-real dynamics transfer and RL with latent-state world models.

## B.1    TRANSFERRING DYNAMICS WITH EXPLICIT REPRESENTATIONS

Sim-to-real transfer of dynamics aims to adapt existing simulators or dynamics models used for planning and policy optimization to better match real-world environments. One way to transfer dynamics from simulation to reality is to calibrate predefined simulator physics parameters to match the target environment, either directly from real data [37] or as a correction to existing parameters [1]. However, doing so can be insufficient if no good approximation of the real environment exists in the space spanned by the allowed range of these parameters. In such cases, a more expressive modification of the simulator state transition function may be needed.

Along these lines, Ball et al. [3] and Arcari et al. [2] calibrate linear error models on simulator transition dynamics using real data for policy adaptation and, respectively, model predictive control. Similarly, Mallasto et al. [28] use affine transport to adapt simulator state dynamics models to real domains. Golemo et al. [10] train an LSTM conditioned on state–action history to predict a state transition residual, and Schperberg et al. [35] efficiently adapt a neural-network state-dynamics

residual by using Unscented Kalman Filtering. Kaufmann et al. [20] employ k-nearest neighbor regression and Gaussian process residuals on transition dynamics and state encodings to calibrate their simulator for drone racing at an expert human level.

Each of these methods relies on the assumption that the environment state can be represented with a compact vector representation with which a generalizable dynamics correction can be learned with a relatively low-complexity model and small real-data requirements. We consider the case where the components of the state are instead in a *high-dimensional* format like images and we do not have a predefined mapping from these states to such a necessary compact vector representation. To adapt simulation transition dynamics under these more difficult conditions, we propose to learn a latent-state world model of the simulation and then train a residual correction on the world model's dynamics to match transitions in the real environment.

### B.2 WORLD MODELS WITH LATENT STATE SPACES

World models [11] with latent state spaces are environment models in which planning and policy learning can be more efficient than with environment states due to a succinct representation of environment states and dynamics. Dreamer [13; 14; 42; 15] models environments in the stochastic POMDP [6] by encoding observations as latent states and reconstructing future latent states, rewards, and observations. The Dreamer architecture allows agents to then train on synthetic experience by rolling out "imagined" trajectories inside of the world model. TD-MPC [17; 16] models deterministic fully observable MDPs by similarly reconstructing future latent states and rewards, as well as task value functions. TD-MPC2 [16] has shown good results learning shared features from a suite of environments to quickly transfer to new ones, while we focus on transferring from a single environment to a similar target environment by avoiding overfitting to limited data.

Concerning exploration with world models, collecting diverse source trajectories was crucial in our experiments for learning transferable features and dynamics. To achieve this, we use Plan2Explore [36], a method compatible with both Dreamer and our proposed DRAW architecture, which trains an auxiliary RL agent alongside the exploit policy to maximize model uncertainty in latent dynamics predictions, promoting wide-reaching exploration.

### B.3 DOMAIN RANDOMIZATION

Domain randomization is widely used for sim-to-real transfer by exposing policies to diverse variations in images [40; 19] or dynamics [34; 29]. In the case of variations with different optimal policies, training on a broad distribution of environment conditions can yield an overly conservative policy. Methods like [39; 7] partially mitigate this issue by leveraging real data to calibrate the parameters of a training-time dynamics domain randomization distribution to more closely represent the target environment.

Similar to the system-parameter-identification methods [37; 1] mentioned in Section B.1, domain randomization relies on having a configurable simulator with parameters that, if set correctly, can sufficiently represent real-world dynamics at training time. Domain randomization thus requires the practitioner to have A) an understanding of which dynamics parameters are likely to be mismatched between simulation and reality, and B) a simulator that allows those specific parameters to be configured. In practice, the nature of disparities between the training-time simulator and adaptation-time real environment may not be known, and the simulator may not be customizable along the necessary parameter dimensions (or at all).

With ReDRAW, we provide a dynamics adaptation method that is agnostic to the types of MDP physics disparities between the source and target environment, however, to do so, ReDRAW relies on being able to zero-shot perception from simulation to real. Given the availability of simulators with high-fidelity visuals like [27; 43; 8] and advances in 3D-reconstruction techniques like neural radiance fields [30] and Gaussian splatting [21] (as we employ in this paper), we believe this can often be a worthwhile tradeoff. In our duckiebots robot sim-to-real experiment, as described in Section 4.2.1, we apply image augmentations and limited camera-parameter randomization to ReDRAW and every baseline. With the exception of the physics domain randomization baseline in Appendix D, we do not vary simulator physics.

### B.4 OFFLINE RL

Offline RL approaches aim to learn well-performing policies from fixed datasets, usually while avoiding taking out of distribution actions that are not well represented in the data [9; 25; 24]. In cases like ours where offline real data is limited but online data from an error-prone simulator is abundant, methods like [31; 32; 18] train on both real and simulated data by down-weighting the effects of updates from simulated transitions with dynamics that differ from the offline real data. However, to train a policy given offline real data, these methods require access to reward real reward labels. Real-reward data can often be difficult to collect on physical robots, and we do not assume access to reward labels in the offline real dataset. Instead, our proposed ReDRAW method learns a reward function conditioned on the current latent state and leverages a fixed latent-state encoding between simulation and the real environment to reuse this reward function in the real-calibrated world model.

## C DUCKIEBOTS EXPERIMENT DETAILS

**Simulation Reward Details**   In simulation, the agent is densely rewarded at each timestep with a value in $[0, 1]$ proportional to its projected velocity along the lane center unless its location is more than 5cm from the lane center, in which case it incurs a penalty of -1. When moving forward, we additionally provide a dense penalty proportional to egocentric yaw velocity to encourage turning while at speed. The simulation episode horizon is 200 steps, slightly more than enough time to complete a lap. We do not provide a termination signal when the horizon is reached. We terminate early with a done signal and a penalty of -100 if the agent drives off the track.

We provide the agent with reward data during simulation pretraining, and we do not provide reward labels in training data collected from the real environment. In order to measure test-time deployment performance, during real evaluation only, we record the robot's location with an HTC Vive motion tracker to measure equivalent simulation rewards, lap times, and the robot's distance from the lane center. Information recorded from the motion tracker is not provided to the agent or world model.

**Image Augmentations**   During simulation pretraining and offline adaptation to real data, we apply image augmentations to world model encoder inputs, but we still train decoder objectives on the original non-augmented images. Figure 6b shows original images (bottom) and their augmented counterparts (top) for both simulation and the real environment offline human demonstration dataset. In world model training for DRAW/ReDRAW and Dreamer, we apply new image augmentations to each mini-batch after it is sampled from the experience buffer.

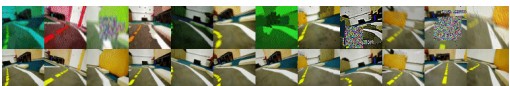 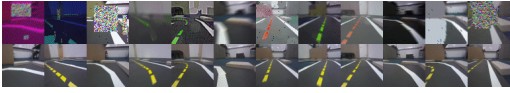

(a) Simulation Images with Augmentations          (b) Real-world Images with Augmentations

Figure 6: Comparison of image observations in simulation and the real world. (**Top**): Augmented images. (**Bottom**): Original images.

## D COMPARISON WITH PHYSICS DOMAIN RANDOMIZATION

We conduct an additional comparison in the Duckiebots domain in which we train DRAW and DreamerV3 with dynamics domain randomization. After pretraining with Plan2Explore in simulation with dynamics domain randomization for 1.4M timesteps, we zero-shot transfer to the target real environments. To represent both the possible range of real robot speeds and the per-step variations in speed present during real-world execution, we vary the simulator's forward and yaw velocity coefficients by a random value each timestep. A coefficient of 1.0 represents the default simulator scaling for forward and yaw velocity. At the start of each episode, for each parameter, we sample a mean scale $\mu_{\text{episode}}^i, i \in \{forward\ vel, yaw\ vel\}$ from $\mathcal{N}(1, 0.1)$. Then, in each timestep, we sample independent per-step parameter scales from $\mathcal{N}(\mu_{\text{episode}}^i, 0.01)$. Additionally, in every timestep, we

Table 3: Mean and SEM performance on the real Duckiebots lane-following task aggregated over 5 episodes each for 4 training seeds. Agents are given 300 steps (30 seconds) to complete a lap from a fixed starting position. *Center Offset* denotes distance from the lane center. Absence of a *Lap Time* indicates all runs either failing to complete a lap or terminating early by driving off the track. In the Duckiebots domain, physics Domain Randomization (DR) underperforms against other zeroshot and transfer-learning approaches.

| Method | Transfer Sim to Unmodified Real | | | Transfer Sim to Actions-Reversed Real | | |
| --- | --- | --- | --- | --- | --- | --- |
| | Avg Dense Reward ($\uparrow$) | Avg Lap Time (sec) ($\downarrow$) | Avg Center Offset ($\downarrow$) | Avg Dense Reward ($\uparrow$) | Avg Lap Time (sec) ($\downarrow$) | Avg Center Offset ($\downarrow$) |
| Dreamer Zeroshot | -1.18 $\pm$ 0.23 | – | 6.86 $\pm$ 0.56 | -2.35 $\pm$ 0.23 | – | 13.36 $\pm$ 1.13 |
| Dreamer DR | -0.19 $\pm$ 0.14 | – | 5.04 $\pm$ 1.56 | -1.53 $\pm$ 0.47 | – | 5.46 $\pm$ 1.64 |
| Dreamer Finetune | -0.87 $\pm$ 0.33 | – | 5.45 $\pm$ 1.55 | -1.61 $\pm$ 0.57 | – | 7.75 $\pm$ 1.53 |
| DRAW Zeroshot | 0.07 $\pm$ 0.06 | **22.41** $\pm$ 0.73 | 5.12 $\pm$ 0.41 | -2.72 $\pm$ 0.42 | – | 9.39 $\pm$ 1.35 |
| DRAW DR | -0.32 $\pm$ 0.12 | **23.75** $\pm$ 1.56 | 8.26 $\pm$ 1.38 | -2.81 $\pm$ 0.33 | – | 7.15 $\pm$ 1.81 |
| ReDRAW (Ours) | **0.38** $\pm$ 0.02 | **22.75** $\pm$ 0.26 | **2.47** $\pm$ 0.26 | **0.39** $\pm$ 0.03 | 24.21 $\pm$ 1.15 | **2.10** $\pm$ 0.39 |

uniformly sample the on-body position and tilt of the robot's camera within a small range of values to simulate oscillations and shocks encountered while driving.

In Table 3, we compare the performance of DRAW and DreamerV3 Domain Randomization (DR) against other methods in the unmodified real and actions-reversed real environments. The unmodified real environment represents a scenario in which the dynamics randomization scheme is well-informed by potential simulator errors. In contrast, the actions-reversed real environment represents a scenario in which a critical disparity between simulation and reality was not anticipated or captured during randomized simulator training.

In the unmodified real environment, DRAW DR achieves low rewards by taking wide turns that cut corners and veer far from the lane center. Because driving off the track results in a large penalty, this behavior can be explained as taking conservative actions. The DRAW DR agent avoids taking otherwise optimal sharp and late turns, likely because this normally optimal behavior can have a dangerous outcome in the randomized simulation given unknown forward and yaw speed coefficients.

Dreamer DR fails in the unmodified real environment with agent behavior in most seeds staying still and occasionally rotating. Shown in Figure 9, Dreamer consistently exhibits instability w.r.t. $M_{sim}$ returns when pretraining with randomized physics in the Duckiebots simulation environment. This behavior was repeatedly seen in preliminary experiments under various randomization conditions both with and without Plan2Explore. These observations suggest a possible limitation in DreamerV3's capacity to learn highly stochastic dynamics, although more experimentation would be needed to fully confirm this.

In the actions-reversed real environment, both DR methods fail to drive because the reversed actions represent an unexpected disparity between simulation and real that was not represented during training.

In both of these real environments, despite not having access to configurable simulator physics, ReDRAW is able to use a small amount of real transition data to calibrate its world model and drive with a near-optimal policy.

## E    LEARNING CURVES DURING PRETRAINING

Figure 7 and Figure 8 show training curves in the source environments in the DMC and Duckiebot domains, respectively. Both DRAW and DreamerV3 converge to similar performance in the source environments. In Appendix H, we also compare ReDRAW with framestacked image components to convey motion in lieu of joint-velocity vectors. The pretraining curves for DRAW with framestacking are similar to the default configuration.

In Appendix D, we evaluate DRAW and Dreamer's zero-shot performance on the real Duckiebots environment after training with physics domain randomization. Figure 9 shows pretraining performance on the simulated environment with episodic physics randomization applied. With physics domain randomization, DreamerV3's source-environment returns decrease to a suboptimal level as training progresses.

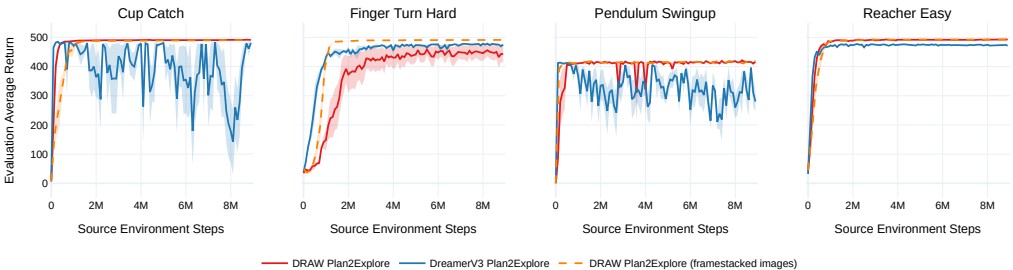

Figure 7: Training curves during pretraining for DRAW and DreamerV3 across four environments from DMC. Plan2Explore is used for data collection during pretraining. The mean and standard error are shown over 4 seeds.

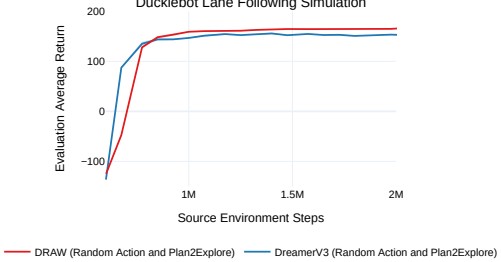

Figure 8: Training curves during pretraining for DRAW and DreamerV3 in the Duckiebot lane following simulation environment. Data collection is performed using random actions for the first 0.6M steps, followed by Plan2Explore for 1.4M steps. Each episode starts from a valid random position. The mean and standard error are shown over 4 seeds.

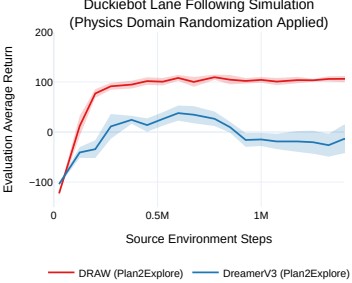

Figure 9: Training curves during pretraining for DRAW and DreamerV3 in the Duckiebot lane following simulation environment with physics domain randomization applied.

## F ARCHITECTURAL ABLATIONS

In this section, we examine how different choices for the inputs of the DRAW forward dynamics function $f_\theta$ and the ReDRAW residual function $\delta_\psi$ affect transfer performance.

### F.1 FORWARD DYNAMICS INPUTS

In the default DRAW architecture, $f_\theta$ is conditioned on the previous latent state $z_{t-1}$, the previous action $a_{t-1}$, and the additional input of the previous latent-state dynamic distribution $\hat{\sigma}_{t-1}$ (or $\hat{\sigma}_{t-1}^{real}$ for ReDRAW). In DMC environments, we compare this choice of inputs against two alternatives: (1) the minimal sufficient set $(z_{t-1}, a_{t-1})$, and (2) conditioning on the encoder latent-state distribution $\sigma_t = q_\theta(z_t|x_t)$. Figure 10a presents the performance of these different dynamics functions on source environments during DRAW Plan2Explore pretraining, while Figure 10b shows their transfer performance on target environments during offline ReDRAW adaptation.

During pretraining, most input choices yield similar source-task performance. However, during adaptation, the default configuration, $f_\theta(z_{t-1}, \hat{\sigma}_{t-1}, a_{t-1})$, consistently outperforms the alternatives, achieving and maintaining higher performance in the target environments. We hypothesize that because including $\hat{\sigma}_{t-1}$ during world model pretraining facilitates gradient propagation over multiple timesteps, this inclusion enables the learning of features that improve long-term predictions.

This advantage is achieved without increasing the residual's complexity, which could have otherwise negatively impacted transfer performance. During ReDRAW adaptation, $\hat{\sigma}_{t-1}^{real}$ serves as an input to $f_\theta(z_{t-1}, \hat{\sigma}_{t-1}^{real}, a_{t-1})$. While conditioning on $\hat{\sigma}_{t-1}^{real}$ increases the dimensionality of $f_\theta$'s input space, it has minimal impact on the complexity of the residual prediction $\delta_\psi$. Since $\hat{\sigma}_t^{real}$ is already an output of the calibrated dynamics,

$$\hat{\sigma}_t^{real} = \text{softmax}(f_\theta(z_{t-1}, \hat{\sigma}_{t-1}^{real}, a_{t-1}) + \delta_\psi(z_{t-1}, a_{t-1})),$$

it can be included as an input to $f_\theta$ without increasing the dimensionality of the input or output spaces of $\delta_\psi$. This helps maintain the residual function's simplicity, reducing the risk of overfitting.

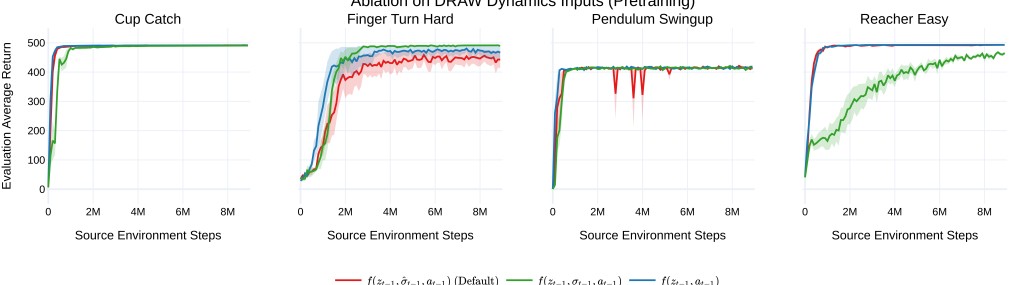

(a) DMC source environment average return during DRAW pretraining with alternate dynamics function inputs.

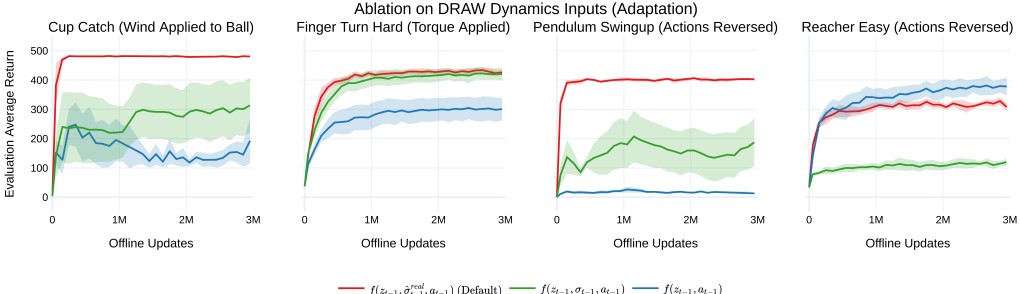

(b) DMC target environment average return during ReDRAW residual adaptation with alternate dynamics function inputs.

Figure 10: Comparison of different dynamics function architectures of DRAW during pretraining (a) and adaptation (b).

### F.2 RESIDUAL INPUTS

Next, in Figure 11, we compare the target environment transfer performance of our default residual function, $\delta_\psi(z_{t-1}, a_{t-1})$, against two alternative input configurations. The first, $\delta_\psi(z_{t-1}, \hat{\sigma}_{t-1}^{real}, a_{t-1})$, conditions on the same inputs as $f_\theta$, while the second, $\delta_\psi(\hat{\sigma}_t, z_{t-1}, a_{t-1})$, additionally incorporates the original source environment dynamics predictions made by the frozen forward belief, $p_\theta(\hat{z}_t | z_{t-1}, \hat{\sigma}_{t-1}^{real}, a_{t-1})$.

Although the additional inputs, $\hat{\sigma}_{t-1}^{real}$ and $\hat{\sigma}_t$, could theoretically provide useful information for the residual prediction task, we observe that their inclusion leads to a decrease in target-environment performance. We hypothesize that conditioning the residual function on an added real-valued vector, alongside the discrete latent-state $z_{t-1}$, significantly expands the space of representable residual functions. Given the limited dataset, this increased complexity likely impairs generalization to the target domain.

This result underscores the importance of bottlenecking state information through the compressed discrete representation $z_t$ for effective low-data adaptation.

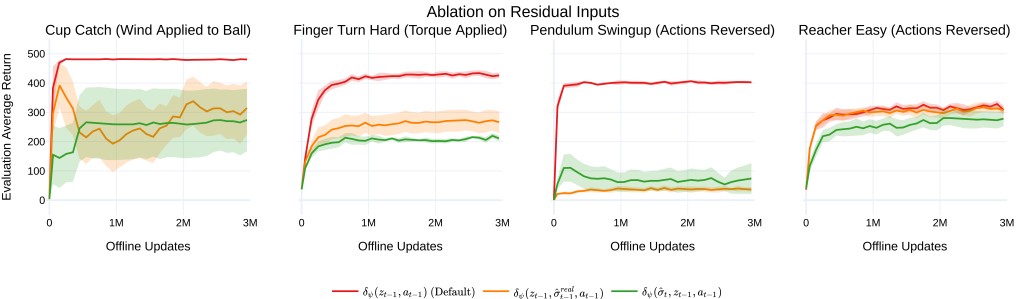

Figure 11: Comparison of different residual inputs for ReDRAW.

## G LATENT RESIDUAL VS NEW DYNAMICS FUNCTION

In this section, we compare the ReDRAW latent-state dynamics residual with an alternative adaptation method that also leverages frozen dynamics predictions learned from the source environment. Specifically, we contrast using a residual with learning a new replacement dynamics function, $g_\psi$, which optionally conditions on the outputs of the original source environment dynamics $f_\theta$. We evaluate three possible definitions for $g_\psi$:

1. $\hat{\sigma}_t^{real} = g_\psi(z_{t-1}, a_{t-1})$, where $g_\psi$ conditions on the same inputs as the ReDRAW residual.

2. $\hat{\sigma}_t^{real} = g_\psi(\hat{\sigma}_t, z_{t-1}, a_{t-1})$, where $g_\psi$ additionally conditions on the frozen DRAW predicted source dynamics distribution, $\hat{\sigma}_t = p_\theta(\hat{z}_t | z_{t-1}, \hat{\sigma}_{t-1}^{real}, a_{t-1})$.

3. $\hat{\sigma}_t^{real} = g_\psi(\hat{z}_t, z_{t-1}, a_{t-1})$, where $g_\psi$ additionally conditions on a discrete latent-state sample from the frozen DRAW source dynamics predictions, $\hat{z}_t \sim \mathrm{MultiCategorical}(\hat{\sigma}_t)$ as in (5).

To train the replacement dynamics function on the offline $M_{real}$ dataset, we employ a dynamics loss term equivalent to (20) used by ReDRAW:

$$\mathcal{L}_g(\psi) = \mathbb{E}_{q_{\bar{\theta}}(z_{1:T}|\zeta^{real})} \left[ \sum_{t=1}^{T} \mathbb{D}[q_{\bar{\theta}}(z_t|x_t)||g_\psi(\hat{z}_t^{real}|\bullet)] \right] \tag{21}$$

Figure 12 presents the average target environment return during adaptation for ReDRAW and all considered replacement dynamics functions. The results show that ReDRAW outperforms all variations of the replacement function baseline, including those that incorporate predictions from the frozen DRAW source dynamics function.

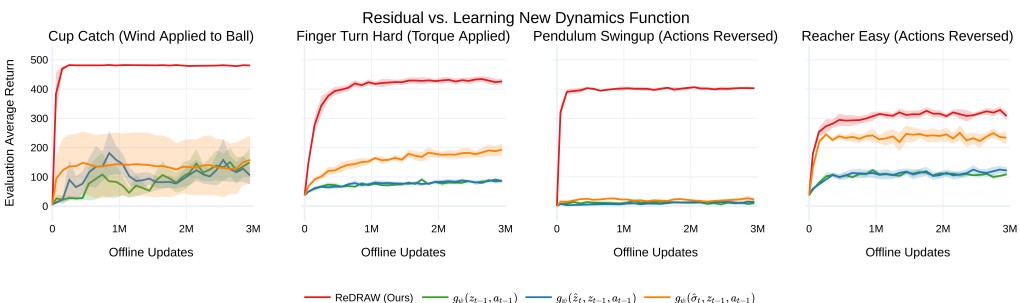

Figure 12: Comparison with a replacement dynamics function $g_\psi$ with the same small capacity as the residual network.

From this experiment, we conclude that the residual operation, which modifies DRAW dynamics predictions without conditioning on them, is a key factor in achieving effective generalization to the target environment.

## H    DMC COMPARISON WITH FRAMESTACKING

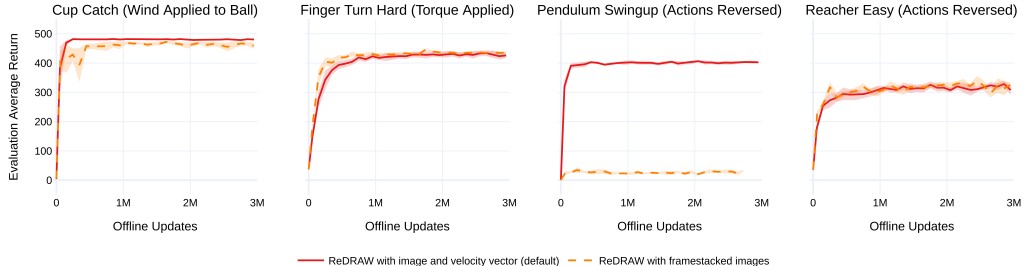

Figure 13: Framestacking images achieves similar transfer performance to a single image + a joint velocity vector in most but not all environments.

In DMC and Duckiebots environments, we ensure full observability by pairing the image state representation with a vector of joint velocities. So long as the time-delta between images remains consistent across source and target environments, framestacking should usually be a viable alternative to convey velocity information. In Figure 13, we compare ReDRAW transfer performance with the default state+velocity vector configuration against framestacking the previous and current image. In all DMC environments except Pendulum Swingup (Actions Reversed), ReDRAW achieves virtually the same performance with either input modality. Curiously, ReDRAW fails to transfer in the framestacked Pendulum Swingup (Actions Reversed) environment despite matching the default method's performance when pretraining in the source environment (Figure 7). Possible causes for this could include Plan2Explore adopting different (and insufficient) source environment data collection strategies with a different state representation, inadequate image fidelity to capture precise velocity behavior with the small target environment dataset, or a more entangled latent-state representation due to decoding a higher-dimensional state. This experiment highlights potential directions for future improvements to ReDRAW.

## I    DMC EXPERIMENT DETAILS

The state spaces for the DMC environments in this work consist of an image of the robot paired with a vector of egocentric joint velocities. We use an action repeat of two, meaning that each episode consists of 500 decision steps, equivalent to 1000 environment steps. Additionally, to preserve state-based rewards, we do not sum rewards over the environment steps skipped due to action repeat.

Below, we describe each pair of source and target environments used in our DMC experiments. The source environment corresponds to the original DMC environment, while each target environment has modified dynamics:

- **Cup Catch**: The agent controls a cup to catch a ball tethered by a string. In the target environment, a constant horizontal wind alters the ball's trajectory, requiring the agent to adapt by compensating for this external force.

- **Finger Turn Hard**: The agent rotates a hinged spinner to a specified goal orientation. In the target environment, an external torque continuously drives the spinner, forcing the agent to counteract this disturbance to maintain control.

- **Pendulum Swingup**: The agent swings a pendulum to an upright position. In the target domain, action effects are reversed, requiring the agent to invert its control policy.

- **Reacher Easy**: The agent maneuvers a two-link arm to reach a target position. As in Pendulum Swingup, actions are inverted in the target environment, posing a challenge for direct policy transfer.

## J  HYPERPARAMETERS

We implement DRAW and ReDRAW code as a modification to the official DreamerV3 implementation [12]. Except where otherwise stated, we use DreamerV3 default hyperparameters for all methods, including a batch size of 16, batch length of 64, and learning rates of $1 \times 10^{-4}$ for the world model and $3 \times 10^{-5}$ for the actor and critic. Additional parameters specific to our method or experiments are listed below.

Table 4: Modified or newly introduced hyperparameters used in experiments.

|  | **Hyperparameter** | **Value** |
|---|---|---|
| all methods | pretraining replay buffer size | 1e7 |
|  | online train ratio | 512 |
|  | Encoder/Decoder CNN Depth | 32 |
|  | Encoder/Decoder MLP hidden layers | 2 |
|  | MLP hidden units | 512 |
|  | image size | $64{\times}64{\times}3$ |
| DRAW/ReDRAW | $K$ (number of categorical distributions) | 256 |
|  | $N$ (number of categorical classes) | 4 |
|  | imagination horizon for actor-critic training | 40 |
|  | $\beta_{pred}$ | 1.0 |
|  | $\beta_{dyn}$ | 1.5 |
|  | $\beta_{rep}$ | 0.5 |
|  | residual learning rate | 1e-2 |
|  | forward dynamics MLP hidden layers | 1 |
|  | residual MLP hidden layers | 1 |
|  | residual MLP hidden units | 256 |

## K  COMPUTE RESOURCES

All experiments were performed on a server with 2x AMD EPYC 7763 64-core processors, 1TB RAM, and 8x NVIDIA RTX A4500 GPUs each with 20GB of VRAM.

Each individual experiment ran on a single GPU. With the exception of Duckiebots per-minibatch image augmentation, which took 30-40 CPU cores, most experiments required less than 8 CPU cores. Plan2Explore pretraining experiments typically ran for 3-6 days, using less than 100GB of RAM, and transfer-learning experiments typically ran for 1-3 days, using less than 30GB of RAM.

## L  LARGE LANGUAGE MODEL USAGE

Large Language Models (LLMs) were used to provide sentence-level editing suggestions while writing this paper.

