# OpenReview forum: "Adapting World Models with Latent-State Dynamics Residuals"
_ICLR.cc/2026/Conference — ICLR 2026 Conference Withdrawn Submission_

### Official Review · Reviewer_wRYx · 2025-10-28

**Soundness:** 3
**Presentation:** 3
**Contribution:** 2
**Rating:** 4
**Confidence:** 3

**Summary:**

The paper addresses the sim-to-real gap in reinforcement learning (RL), where policies trained in simulation often fail in real-world environments due to mismatched dynamics. The authors propose ReDRAW, a method that learns residual corrections in the latent state space of a pretrained world model instead of directly modifying high-dimensional observed states. They first train a discrete latent-state world model, DRAW, in simulation, then freeze its parameters and calibrate it using a small, reward-free real-world dataset by learning residuals that correct latent dynamics. This allows RL agents to be trained through imagined rollouts under corrected dynamics and then deployed in real environments. Experiments on DeepMind Control Suite tasks and a Duckiebot sim-to-real transfer demonstrate that ReDRAW adapts effectively to new dynamics and avoids overfitting with limited real data.

**Strengths:**

- The experiments include both synthetic (DeepMind Control Suite) and physical (Duckiebot) experiments, showing consistent improvements in performance and robustness.
- The method achieves effective adaptation using only reward-free datasets and demonstrates superior stability compared to DreamerV3 and fine-tuning baselines.

**Weaknesses:**

- The experiments mainly cover moderate and structured physics changes (e.g., action reversal, added torque); it’s unclear how well the method scales to highly nonlinear or discontinuous real-world discrepancies.
- The residual correction network is intentionally low-complexity, which could limit the capacity to model complex dynamic shifts.
- The method assumes fully observable environments, limiting applicability to partially observable or stochastic domains.
- The proposed world model DRAW only has minor modifications to Dreamer.

**Questions:**

- How does the data quality of offline datasets affect the final performance?
- What does "Online Exploit Exploration" refer to in Figure 3?

---

### Official Review · Reviewer_e8iZ · 2025-10-30

**Soundness:** 2
**Presentation:** 3
**Contribution:** 3
**Rating:** 4
**Confidence:** 4

**Summary:**

This paper introduces a new architecture for model-based reinforcement learning. The model can be pre-trained in simulation and fine-tuned using a small amount of real-world data. Fine-tuning is done by learning a correction model that shifts the latent state logits to match the real-world transition dynamics.

**Strengths:**

**Novelty**

The approach to learn the deviation to the real-world dynamics has been explored for improving simulation, but not as much in MBRL. The depicted architecture seems to be sound and its components are thoroughly ablated.

**Experiment on physical hardware**

The experiment on real-world hardware is a strong indicator for the method's soundness.

**Weaknesses:**

My main concern with this paper is the restriction to fully observable settings, despite the focus on high-dimensional and noisy inputs such as images. In a fully observable MDP, it is unclear why one would choose to include an observation reconstruction model that predicts images. Moreover, if the full state can already be reconstructed, it is not evident why learning the reward function would be advantageous, rather than simply computing rewards directly from the predicted state.

**Questions:**

What is the reason that you did not implement the correction method on top of Dreamer's original RSSM, allowing for environments with partial observability?

---

### Official Review · Reviewer_rEUG · 2025-11-01

**Soundness:** 2
**Presentation:** 3
**Contribution:** 2
**Rating:** 4
**Confidence:** 4

**Summary:**

This paper aims to tackle the sim-to-real gap in RL for high-dimensional observations with limited real data. It proposes ReDRAW, which pretrains a discrete latent-state autoregressive world model (DRAW) on simulation data, then freezes DRAW and trains a lightweight residual module on a small real-world dataset to correct latent-state dynamics. ReDRAW enables imagined rollouts in the corrected latent space and is validated on four vision-based DMC tasks and a real Duckiebot lane-following task.

**Strengths:**

1. The paper proposes a well-designed model. ReDRAW shifts dynamics correction from high-dimensional observations to a discrete latent space, avoiding manual feature engineering and overfitting in low-data regimes.
2. The method is evaluated on both controlled DMC tasks and real Duckiebot data, showing robust real-world performance. Ablations validate the effectiveness of key architectural choices.
3. The paper is well-structured and clearly written.

**Weaknesses:**

1. My key concern is about the modest innovation in world model architecture and RL algorithms. ReDRAW’s core model-based RL (MBRL) framework largely mirrors DreamerV3, with the main modification being the addition of a residual dynamics module. The architectural contribution feels incremental rather than paradigm-shifting.
2. ReDRAW’s frozen encoder assumes high similarity between sim and real environments, especially in visual features and task definitions. Zero-shot perception transfer via image augmentation/camera randomization only works if sim and real images share task-relevant features. Large visual gaps or task differences (e.g., sim grasps objects while real pushes them) could render the latent representation ineffective, limiting its applicability across diverse scenarios.
3. In the experiments, the model is compared against a limited set of baselines, omitting several state-of-the-art sim-to-real methods. Including these comparisons would better contextualize ReDRAW’s advantages, particularly its ability to adapt with minimal real-world data in high-dimensional tasks.

**Questions:**

How does ReDRAW perform under large visual disparities? ReDRAW uses image augmentation and camera randomization to handle minor visual gaps, but extreme visual differences may break the frozen encoder. Could a lightweight visual adapter trained on real images preserve the low-data advantage while improving robustness? Testing this would address practical limitations in sim-real similarity.

---

### Official Review · Reviewer_zhHS · 2025-11-01

**Soundness:** 3
**Presentation:** 3
**Contribution:** 2
**Rating:** 8
**Confidence:** 3

**Summary:**

This paper proposes ReDRAW, a domain adaptation method designed for pixel-based sim2real MBRL. It learns an additional residual function upon transition function learnt in simulator, rectifying the discrepancy between simulator and target domains. Empirical results on DMC together with a robotic control task Duckietown demonstrate that ReDRAW maintains its  performance  during target finetune, while zero-shot or finetuning DV3 fails due to overfitting on a small data regime.

**Strengths:**

1. It's important to study domain adaptation with residual transfer in a pixel-based MBRL setting. This subject is valuable especially for target tasks with image observations and yet to determine rewards.

2. The method proposed demonstrates its capability of maintaining performance during target domain finetuning, while directly finetune DV3 suffers from overfitting.

3. The paper is well-structed and easy to follow.

**Weaknesses:**

1. From my perspective, it would be helpful to report the performance of pretrained DRAW in its source task, which improves the understanding of the domain gap between source and target domain.

2. Although ReDRAW is able to maintain its performance during target training, it shows marginal improvement upon a Dreamer Finetune with early stopping.

**Questions:**

While overall this paper is good, following questions would strengthen my understanding of the method proposed.

1. Is it necessary to remove deterministic $h_t$ from the latent state space? What if $h_t$ is kept while the residual $e_t$ is added to the stochastic part $z_t$?

2. Related to W1, how does ReDRAW perform when the domain discrepancy becomes larger? Foe example, what if the velocity of the wind becomes larger in the Cup Catch task?

---

### Note · Authors · 2025-11-12

I have read and agree with the venue's withdrawal policy on behalf of myself and my co-authors.